# Why does the X chromosome lag behind autosomes in GWAS findings?

**Ivan P. Gorlov** [ID] *, **Christopher I. Amos**

Baylor College of Medicine, Institute for Clinical & Translational Research, One Baylor Plaza, Houston, Texas, United States of America

* ivan.gorlov@bcm.edu

## Abstract

The X-chromosome is among the largest human chromosomes. It differs from autosomes by a number of important features including hemizygosity in males, an almost complete inactivation of one copy in females, and unique patterns of recombination. We used data from the Catalog of Published Genome Wide Association Studies to compare densities of the GWAS-detected SNPs on the X-chromosome and autosomes. The density of GWAS-detected SNPs on the X-chromosome is 6-fold lower compared to the density of the GWAS-detected SNPs on autosomes. Differences between the X-chromosome and autosomes cannot be explained by differences in the overall SNP density, lower X-chromosome coverage by genotyping platforms or low call rate of X-chromosomal SNPs. Similar differences in the density of GWAS-detected SNPs were found in female-only GWASs (e.g. ovarian cancer GWASs). We hypothesized that the lower density of GWAS-detected SNPs on the X-chromosome compared to autosomes is not a result of a methodological bias, e.g. differences in coverage or call rates, but has a real underlying biological reason–a lower density of functional SNPs on the X-chromosome versus autosomes. This hypothesis is supported by the observation that (i) the overall SNP density of X-chromosome is lower compared to the SNP density on autosomes and that (ii) the density of genic SNPs on the X-chromosome is lower compared to autosomes while densities of intergenic SNPs are similar.

## Author summary

One of the most striking observations from the Genome Wide Association Studies (GWAS) is that the density of GWAS hits is much lower on X-chromosome compared to autosomes. This was initially explained by technical/analytical reasons such as lower coverage and lack of adequate methods to analyze X-chromosomal SNPs. Since then, a better coverage and better analytical methods to analyze X-chromosomal SNPs were developed. We recently revisited the issue and found that the density of GWAS hits on X-chromosome is at least 5-fold lower compared to autosomes. We demonstrated that the difference cannot be explained by technical or analytical reasons. We proposed a hypothesis of a real biological phenomenon underlying X versus autosomal differences in the density of GWAS-detected SNPs, namely that X-chromosome has a lower density of functional

**Data Availability Statement:** The data analyzed herein was previously published and openly available. All data relevant to our analysis are shown in supplementary materials. We use GWAS-detected SNPs reported in the Catalog Of Published Genome Wide Association studies https://www.ebi.

ac.uk/gwas/. To count number of known SNPs on individual chromosomes we used data from dbSNP database https://www.ncbi.nlm.nih.gov/snp/, one thousand genome database https://www.genome.gov/27528684/1000-genomes-project#al-3, and TOPMed project https://topmed.nhlbi.nih.gov/. Sizes of intronic, exonic intergenic regions for individual chromosome were estimated using data from the consensus coding sequence (CCDS) database https://www.ncbi.nlm.nih.gov/projects/CCDS/CcdsBrowse.cgi.

**Funding:** Partial financial support was received from National Institutes of Health grants U19CA203654, U19CA203654S1 and Cancer Prevention and Research Institute of Texas grant RR170048. The funders had no role in study design, data collection and analysis, decision to publish, or preparation of the manuscript.

**Competing interests:** The authors have declared that no competing interests exist.

polymorphisms compared to autosomes because of a stronger selection against X-chromosomal mutations since X-chromosomal variants are more exposed to natural selection due to hemizygosity in males and X-chromosome inactivation in females. The hypothesis is supported by the analysis of the densities of intergenic, intronic and exonic SNPs on human chromosomes.

## Introduction

The X-chromosome comprises about 156 million base pairs, which is comparable to the size of chromosome 7. For most of the X-chromosome, crossovers are limited to females only, resulting in a stronger linkage disequilibrium on the X-chromosome compared to autosomes [1]. The X-chromosome has two small pseudoautosomal regions (PARs) on Xp and Xq. While genes in PAR regions are expressed from both homologs in females, X-inactivation for most of the X chromosome that does not have a Y complement occurs during embryogenesis. In males, recombination between the X and Y chromosomes is limited to the pseudoautosomal regions which display an enormous excess of recombination compared to the genome average [2]. Both positive and negative (purifying) selection are stronger for the X chromosomal mutations because hemizygosity in males and X-chromosome inactivation in females make them more open to selective pressure compared to mutations occurring on diploid autosomes [3,4].

Genome-wide association studies (GWASs) have advanced the understanding of genetic control of human diseases and phenotypes [5,6]. Thousands of associations between Single Nucleotide Polymorphisms (SNPs) and phenotypic traits/diseases have been reported [7–10]. Since the X-chromosome has a number of unique characteristics compared to autosomes, it is important to understand why there are differences between the X-chromosome and autosomes in terms of GWAS findings.

A review of the distribution of GWAS hits by chromosomes found a substantial deficit of GWAS-detected SNPs on X-chromosome compared to autosomes (see study by Wise at al. [11]). The initial explanation of this deficit was underutilization of X-chromosomal SNPs by GWAS researchers because of lack of effective methods to analyze X chromosomal SNPs. However X-chromosome specific methods for GWAS were published 15 years ago [12,13] and since then it has been an active area of research [14,15]. Despite availability and ever improving analytical tools for X-chromosomal SNPs the shortage of GWAS SNPs located on X-chromosome persisted.

We reviewed summary statistics of thousands of published GWASs to identify proportions of SNPs associated with diseases/phenotypes for all chromosomes separately. We found that the proportion of disease/phenotype-associated SNPs is much lower for the X chromosome compared to autosomes. The persistent difference between X-chromosomal and autosomal SNPs in the densities of GWAS hits suggest that it is not a result of a "neglect" of X-chromosome by GWAS researchers [16] but that the deficit has a real underlying biological basis. We propose that a lower density of functional SNPs underlies the deficit of GWAS hits on X-chromosome compared to autosomes.

## Results

### Overall SNP densities on individual chromosomes

Fig 1 shows the overall density of SNPs computed as the number of SNPs per kilobase. SNPs from 3 sources are shown separately from the left to right: NCBI dbSNP database, 1000 Genome Project, and TOPMed project. Density of SNPs on X-chromosome is somewhat

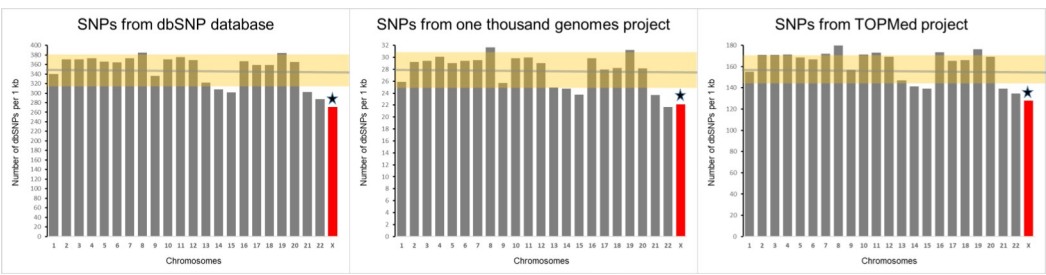

**Fig 1. A comparison of X-chromosome and autosome by the overall SNP density.** Green horizontal line depicts the mean for autosomes. Highlighted area represents SD for autosomes. Star marks the significant difference between X-chromosome and the mean for autosomes. The density was estimated as the number of SNPs per kilobase of chromosomal sequence.

lower compared to the average density on autosomes. In all 3 cases the differences between X-chromosome and autosomes in the density of SNPs were statistically significant with the corresponding p-values being: p = 0.003, 0.02 and 0.006.

## The densities of the GWAS-detected SNPs on individual chromosomes

We found that the number of GWAS-detected SNPs per megabase is 6.8 times lower for X-chromosome compared to the average density for autosomes– 8.7 SNPs for X-chromosome compared to 59.4±2.9 for autosomes (Z-score = 3.8, p = 0.0001) (Fig 2, left panel). Accounting for the differences in the overall SNP density between X-chromosome and autosomes (see Fig 1) does not change the result materially (right panel of Fig 2). The average number of GWAS-detected SNPs per thousand SNPs reported by the NCBI dbSNP database is 0.17±0.01; however, the number of GWAS-detected SNPs per thousand X-chromosome SNPs is 0.03 (Z-score = 3.9, p = 0.00004). Therefore, there is a 5.7-fold difference between autosomes and X-chromosome in the proportion of GWAS-detected SNPs among all SNPs.

## Coverage of X-chromosome and autosomes by common genotyping platforms

We analyzed chromosome coverage for 28 most often used genotyping platforms to test if the lower number of GWAS-detected SNPs on X-chromosome is a result of a lower coverage of X-

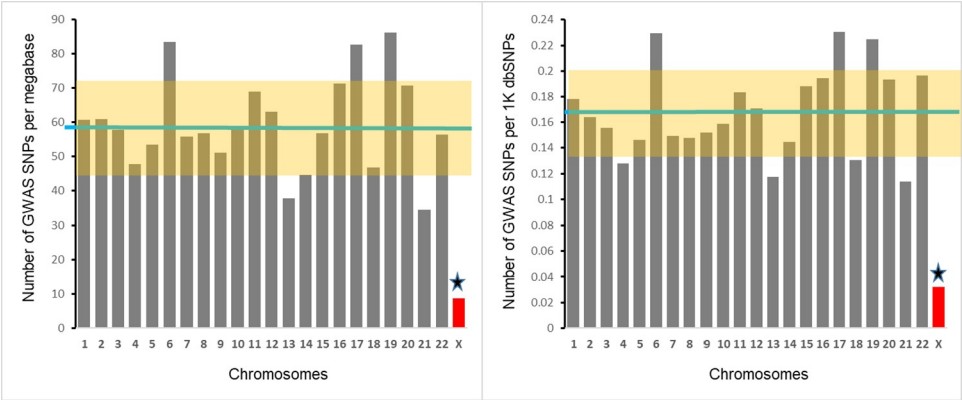

**Fig 2. Chromosomal distributions of density (left panel) and fraction (right panel) of GWAS-detected SNPs.** Left panel depicts the number of GWAS-detected SNPs per megabase, right panel shows the number of GWAS-detected SNPs per thousand SNPs. Each bar represents a chromosome. Green horizontal line represents the mean for all autosomes. Highlighted area represents SD for autosomes. Star marks the significant difference between X-chromosome and the mean for autosomes.

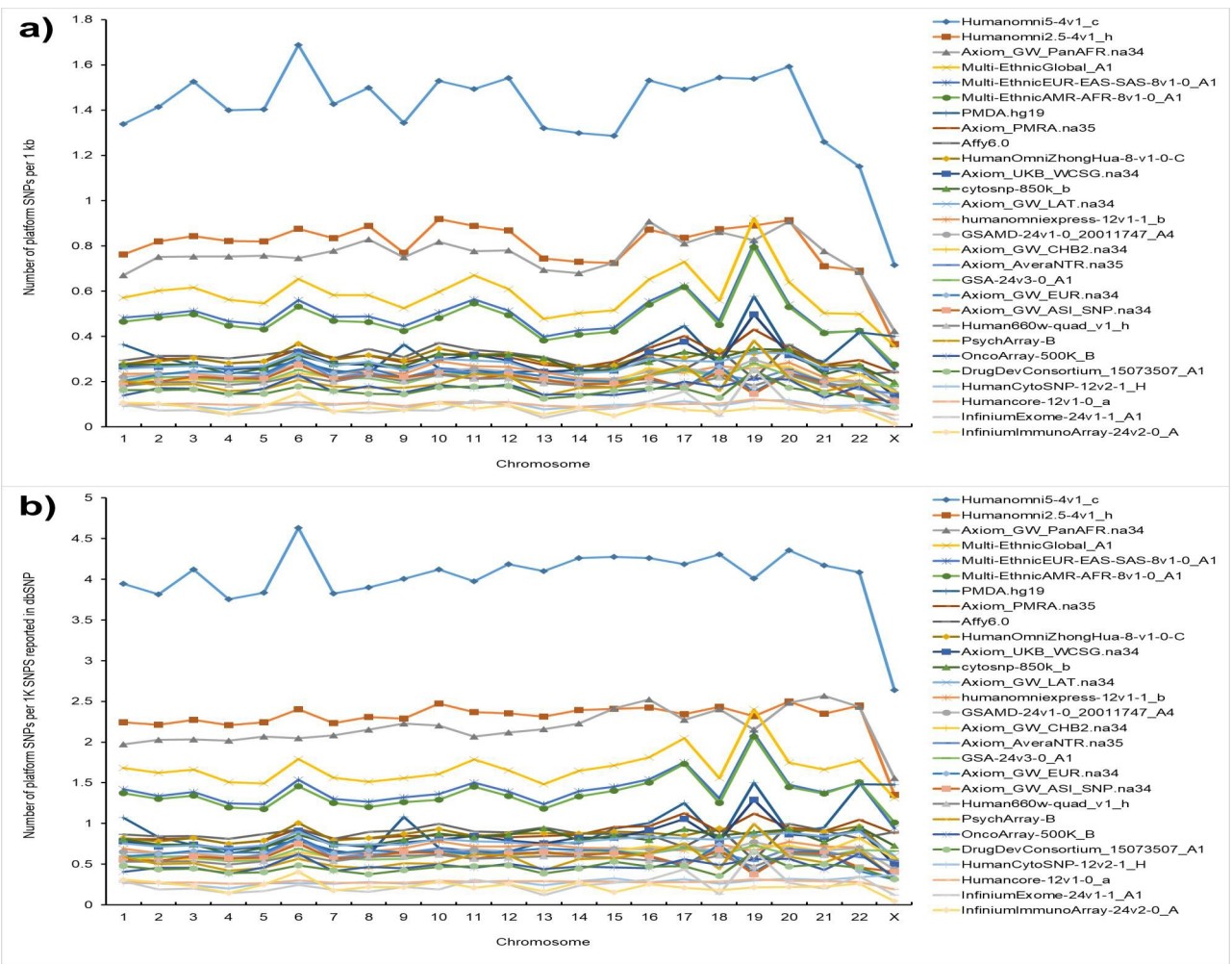

**Fig 3. SNP coverage of individual human chromosomes on common genotyping platforms. (a)** The number of SNPs on the platform per 1kb of the chromosomal sequence. **(b)** The number of SNPs on the genotyping platform per one thousand of the SNPs reported by dbSNP database for a given chromosome.

chromosome by genotyping arrays. The number of SNPs on individual chromosomes for each of the 28 platforms can be found in S2 Table. We used two metrics to test if all chromosomes are equally represented on genotyping platforms: (i) the number of platform SNPs per 1,000 nucleotides (chromosomal density); and (ii) the number of platform SNPs per 1,000 dbSNP SNPs reported for a given chromosome (fraction of genotyped SNPs). Fig 3 shows the results of the analysis. The results for these two metrics were similar. We found a lower density of X-chromosome SNPs compared to the SNPs on autosomes (upper panel of Fig 3): the mean ratio of the X-chromosome to autosome coverage across all platforms was 0.59±0.04, which significantly deviates from 1 expected under the assumption of equal coverage of X-chromosome and autosomes (t-test = 14.0, df = 28, p = 6.8x10$^{-14}$). When the fraction of genotyped SNPs was used as a metrics for the coverage, the mean ratio of the X-chromosome to autosome coverage across all platforms was 0.75±0.05, which also significantly deviates from 1 expected under assumption of equal coverage of X-chromosome and autosomes (t-test = 15.0, df = 28, p = 3.8x10$^{-9}$). Therefore, coverage of X-chromosomal SNPs is 25–40% lower compared to the coverage of autosomal SNPs.

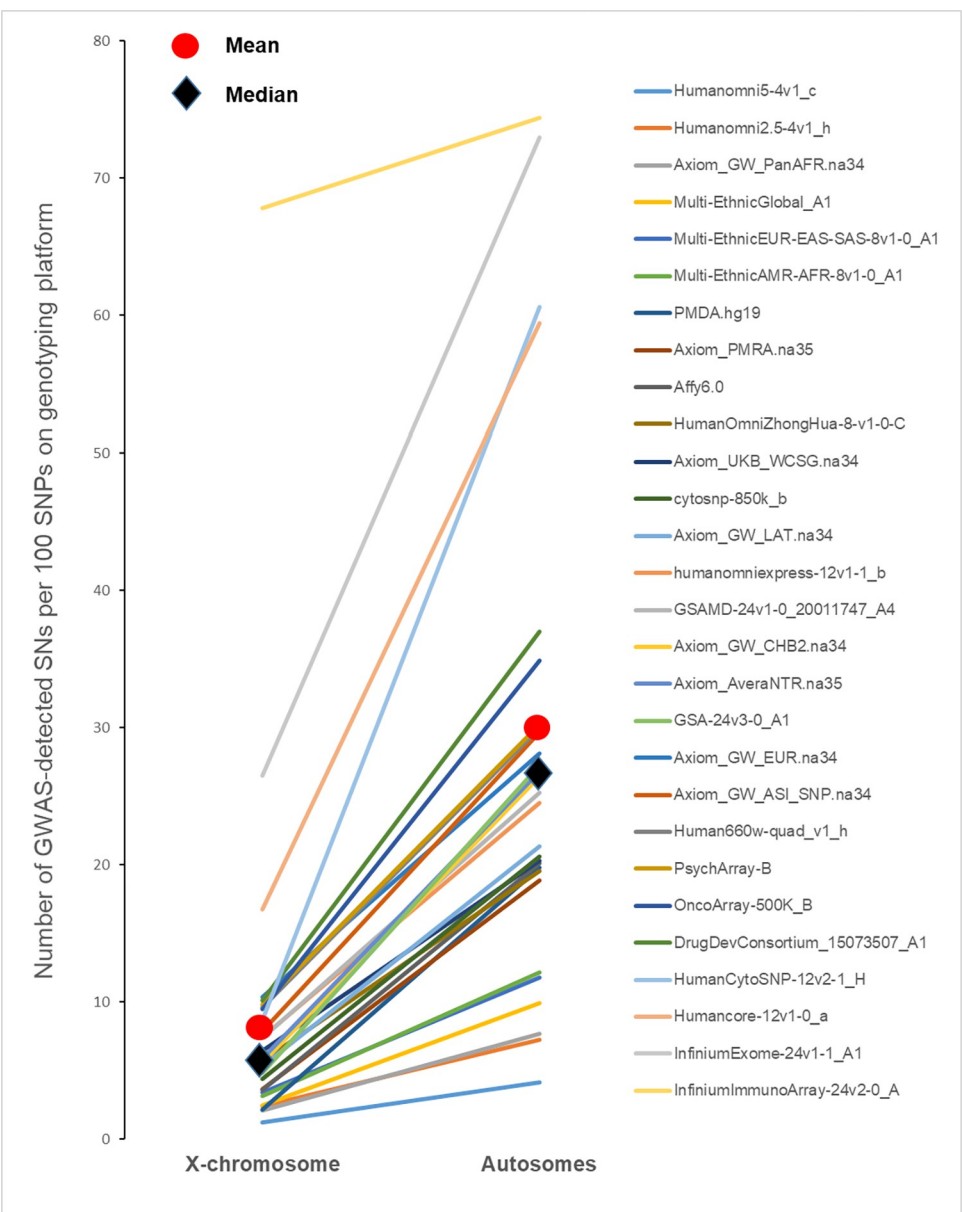

**Fig 4. The number of GWAS-detected SNPs on X-chromosome and autosomes per 100 SNPs on genotyping platforms.**

## The number of GWAS-detected SNPs per hundred SNPs on genotyping platform

To test if the lower density GWAS hits can be explained by the lower coverage of SNPs located on X-chromosome we compared the densities of GWAS hits after adjustment for chromosome coverage. Our goal was to compare density of GWAS-detected SNPs among SNPs that were actually genotyped. We retrieved lists of SNPs for each genotyping platform and among them identified SNPs ever reported by GWASs. To make the results comparable across chromosomes we counted the number of GWAS-detected SNPs per 100 SNPs (Fig 4, S3 Table). The goal of the Fig 4 (S3 Table) was to demonstrate that the difference between autosomal and X-

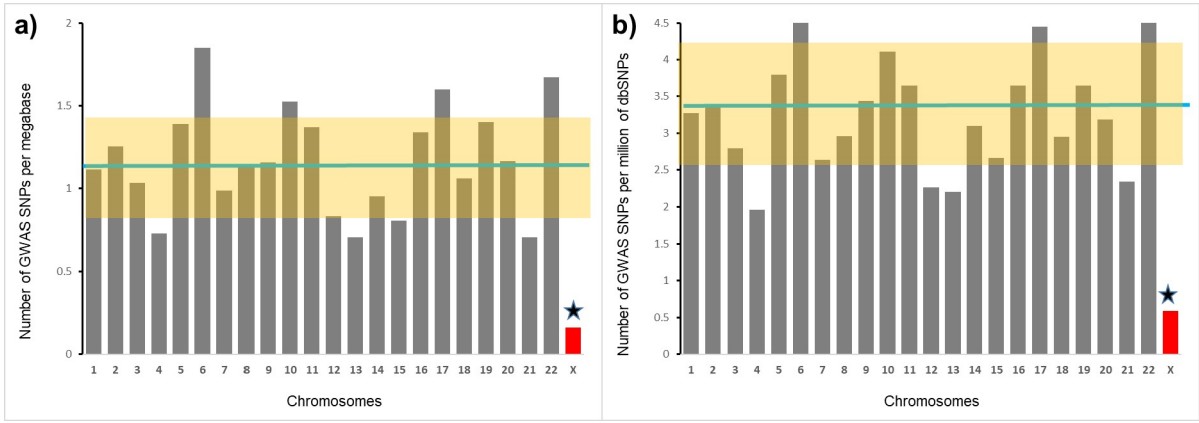

**Fig 5. A comparison of X-chromosome and autosomes by the density of GWAS-detected SNPs in the female-only GWASs.** The green horizontal line is the average among all autosomes; the highlighted area represents SD for autosomes; star marks a significant difference between X-chromosome and the average for autosomes. **(a)** The number of female-restricted GWAS-detected SNPs per megabase. **(b)** The number of female-restricted GWAS-detected SNPs per million of SNPs reported by dbSNP database for a given chromosome.

chromosome SNPs in the density of GWAS hits remains when we consider SNPs that **were actually included in GWASs**. Mean and median number of SNPs reported in at least one GWAS for X-chromosomal SNPs were 9.1% and 5.7% correspondingly, and 27.9% and 24.9% for autosomal SNPs. For each platform we also computed the ratio of the success rate for autosomal SNPs to the success rate for X-chromosomal SNPs. The mean ratio across 28 platforms was 4.03±0.28 (sign test = 5.1, p = 0.0000003). Therefore, autosomal SNPs have a much higher probability to be associated with a phenotype or disease compared to SNPs located on the X-chromosome after taking into account a lower X-chromosome coverage.

## SNPs detected in female-restricted GWASs

Because of hemizygosity the number of genotyped X-chromosomes in males is half of the number of X-chromosomes in females compared to females while the number of genotyped autosomal SNPs is the same. This difference leads to a smaller effective sample size in males and, therefore, may affect the statistical power. Female-only GWASs, however, are expected to have a similar statistical power for X-chromosomal- and autosomal SNPs. The complete list of female-restricted diseases and phenotypes with the number of GWAS hits are shown in S4 Table. In total, 163 female-only phenotypes were used in the analysis, with the total number of GWAS hits of 4,581, comprising about 3% of the total number of SNPs reported in the Catalog of Published GWASs.

The results from the analysis are essentially identical to the results from all-GWAS-hits-together analysis (Fig 5). In female-restricted GWASs, the average number of GWAS-detected SNPs per megabase was 59.84±3.14 for autosomes, while the number of GWAS-detected SNPs per megabase of X-chromosomal sequence was much lower—8.69 (Z-score = 3.5, p = 0.0003). The average number of GWAS-detected SNPs per million of autosomal SNPs was 3.33±0.2 while the number of GWAS-detected SNPs per million of reported X-chromosomal SNPs was 0.59 (Z-score = 2.9, p = 0.002). For female-restricted phenotypes there is a 6.9-fold difference between autosomes and X-chromosome in the density of GWAS-detected SNPs and there is a 5.6-fold difference in the proportion of GWAS-detected SNPs among all reported SNPs. These differences are very similar to the all-GWAS-hits-together analysis: 6.8-fold and 5.7-fold differences, correspondingly (see Fig 2).

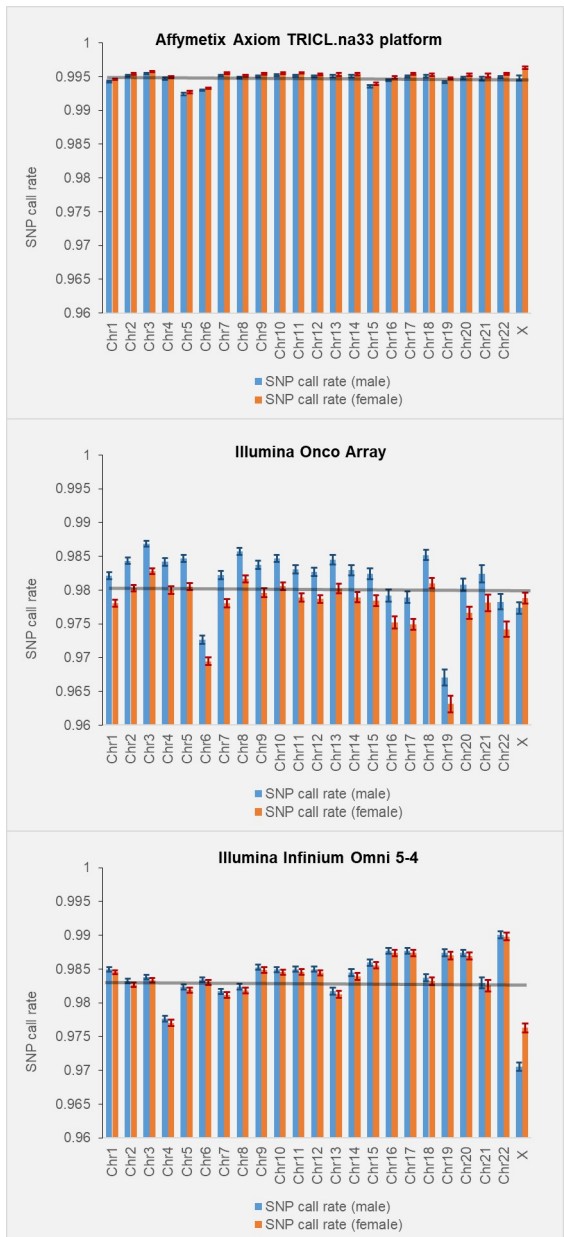

**Fig 6. SNP call rates for the SNPs stratified by chromosome.** Call rates in males and females are computed and shown separately. Error bars indicate standard errors (SEs).

## Call rates for X-chromosomal and autosomal SNPs

One of the suggested "technical" explanations of the low density of GWAS hits on X-chromosome states that ". . .there might be problems with genotype calling for hemizygous males as a result of the lower intensity of some X chromosome variants, and so such males might cluster differently than females." [11]. To test this conjecture we revisited call rate data from several GWASs we have conducted or participated in [17–20]. Call rates were estimated for individual chromosomes and stratified by gender (Fig 6). For Affymetrix and OncoArray, call rates for X-chromosomal SNPs did not differ from the average rate call for autosomes. For Infinium

Omni array the call rate for X-chromosomal SNPs was slightly lower compared to average for autosomes: 0.971±0.002 versus 0.984±0.001. It is hard to say if this specifically relates to X-chromosome or it is a part of global inter-chromosomal differences in call rates, for example chromosome 4 also shows a slightly lower call rate compared to other autosomes. In any case, 1–2% differences in the SNP call rates cannot explain 6-fold differences in the density of GWAS-detected SNPs. The call rates were similar between autosomal and X-chromosomal SNPs both in males and females on other genotyping platforms also (personal report from Dr. Doheny (Johns Hopkins)).

## Testing X-chromosomal SNPs for Hardy Weinberg Equilibrium

Testing SNPs for deviation from Hardy Weinberg Equilibrium (HWE) is a common QC test for genotyping quality. Including males in HWE test for X-chromosomal SNPs may lead to their preferential exclusion from the analysis because X-chromosomal SNPs are frequently deviate from HWE due to hemizygosity. The Initially recommended approach to deal with this issue is to run HWE test only in females [21–23]. More recently, an exact statistical test based on conditional distribution of the number of heterozygotes given the minor allele count has been proposed [24]. The test takes both male and female genotypes into account. For some SNPs the results from female-restricted HWE testing versus all-individuals-included testing are different, but overall the differences are not substantial (see Fig 7 in [24].) We randomly reviewed 20 GWAS studies published in the last 5 year and found none including males in HWE testing. The list of the GWASes randomly selected to test if they indicated including males in HWE testing can be found in S5 Table.

## The density of genic SNPs is lower on X-chromosome compared to autosomes, while the density of intergenic SNPs is similar on X-chromosome and autosomes

We hypothesized that the density of functional SNPs is higher on autosomes compared to X-chromosome. To test this hypothesis we stratified dbSNP SNPs into those located in genic and intergenic regions and estimated their densities for each chromosome separately. SNPs located in genic regions are more likely to be functional compared to intergenic SNPs [25]. The latter is supported by the observation that SNPs in genic regions explain more phenotypic variation [26] and have a higher GWAS reproducibility rate compared to intergenic SNPs [25]. The density of SNPs located in genic regions of autosomes was higher than that of X-chromosome (Z score = 8.2, $P = 8.7 \times 10^{-16}$), while the density of intergenic SNPs was similar for X-chromosome and autosomes (Z score = 1.2, P = 0.20).

Since the dbSNP database includes SNPs identified by targeted sequencing, this potentially can lead to a bias–a higher SNP density in more frequently targeted regions. Also because the data provided to the dbSNP database are submitted by individual users, in some cases this can lead to a bias against X-chromosome SNPs when the authors use inadequate variant calling methods [27]. To deal with this issue we have conducted an analysis using only SNPs detected by whole genome sequencing: SNPs from the 1000 Genomes Project (1KG) and SNPs from TopMED database. Fig 7 shows the densities of SNPs in intergenic, intronic and exonic regions separately for all autosomes and X-chromosome. Both 1KG and TOPMed SNPs' densities in intergenic regions were similar on X-chromosome and autosomes. For 1KG, SNP densities of intergenic SNPs were 2.7±0.1 and 2.2, respectively (p = 0.11). For intronic SNPs, the densities for autosomes and X-chromosome were 3.0±0.1 and 2.1 correspondingly ($p = 1.2 \times 10^{-6}$). For exonic SNPs, the difference between autosomes and X-chromosome was

## 1000 Genome Project SNPs

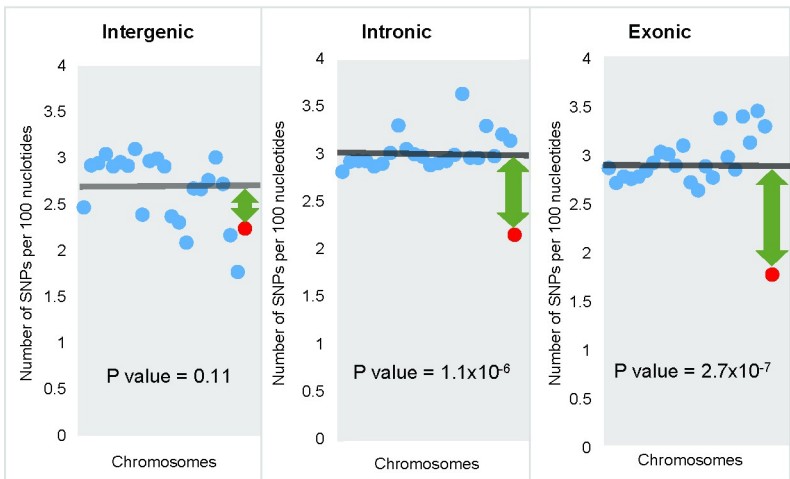

## TOPMed Project SNPs

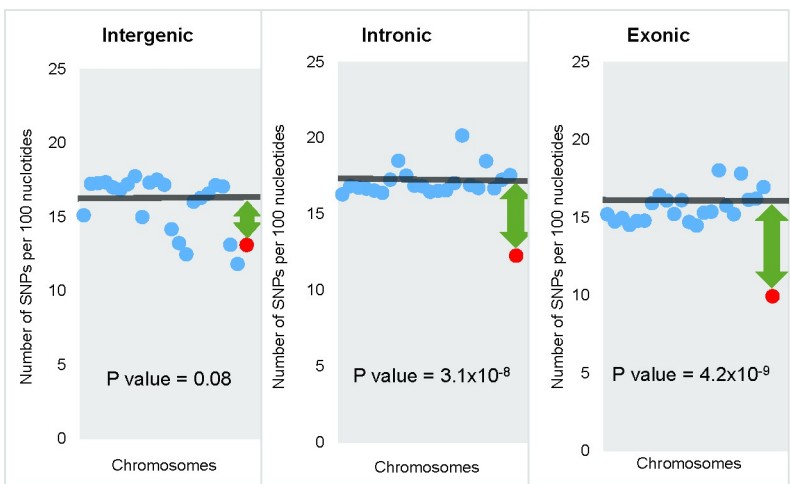

**Fig 7. The densities of 1KG and TOPMed SNPs in intergenic, intronic and exonic regions.** Each dot represents a chromosome: blue dots–autosomes and red dot–X- chromosome. Horizontal lines show the mean SNP density for all autosomes. Green arrows depict the difference between the mean autosomal and X-chromosome densities. Positions of chromosomes on the Fig from left to right correspond to chromosomal numbers.

slightly higher: 3.0±0.1 and 1.8, correspondingly (p = $2.2 \times 10^{-7}$). The results for TOPMed SNPs were similar to the results for 1KG SNPs (lower panel of Fig 7).

## Proportions on segregating sites for missense mutations stratified by predicted effect on protein function

The ratio of segregating sites to the number of potential sites was used as a measure of purifying selection [28]. Sites under strong selection are expected to be less polymorphic. The results of the analysis stratified by chromosome are shown on Fig 8. Proportions of segregating sites increase with a higher Envision score. There are striking differences in the proportion of segregating sites between X-chromosome and autosomes: X-chromosome has a consistently lower proportion of segregating sites compare to autosomes. This is consistent with the idea that selection against X-chromosome missense mutations is stronger compared to autosomal missenses mutations.

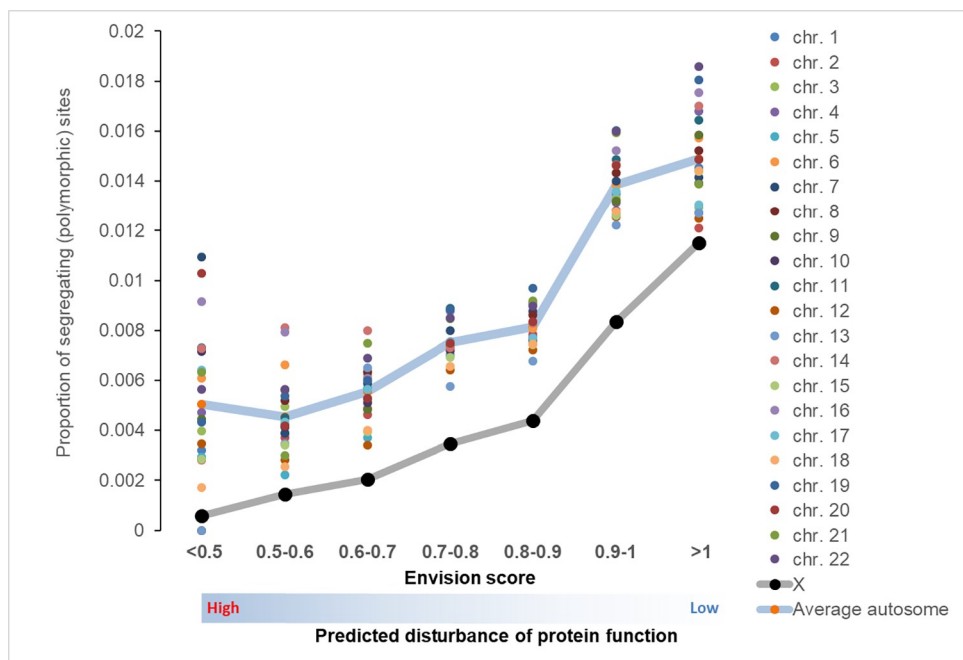

**Fig 8. The proportion of segregating sites for missense mutations categorized by Envision score.** The lower envision score is associated with a stronger effect on protein function. Of note, there is a higher inter chromosomal variation in the proportions of segregating sites on the tails of the distribution because of small sample sizes for those categories (see S6 Table).

## Pseudoautosomal region 1 (PAR1)

We looked at the distribution of SNPs along X-chromosome (Fig 9). We binned the X-chromosome into 52 3-megabase segments. We chose this segment size because it is close to the size of the PAR1 region which is 2,781,479 bases. The total number of dbSNP SNPs in PAR1 is 63,848, which is almost 20 times lower compared to the average number of SNPs in other regions of X-chromosome– 828,112. A total of 15 GWAS-detected SNPs are located in PAR1,

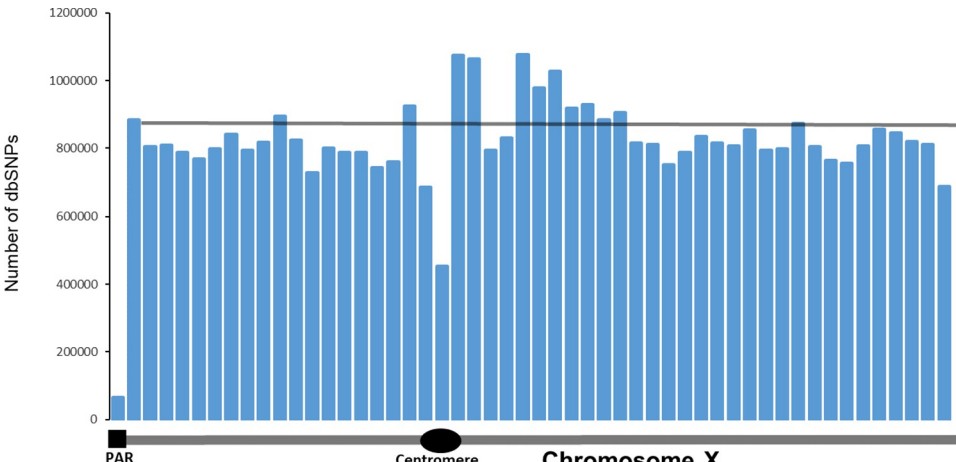

**Fig 9. The distribution of the number of SNPs reported in dbSNP database.** Each bar represents a 3-megabase fragment of X-chromosome. The first bar corresponds to the position and size of the pseudoautosomal region 1. Thin gray horizontal line represents the average number of SNPs in all bars except PAR1.

which translates into 6.7 SNPs per megabase. The estimated number of GWAS-detected SNPs per thousand of dbSNP variants in PAR1 is 0.21, which is comparable to the density of GWAS detected SNPs on autosomes– 0.17. Therefore, the fraction of GWAS-detected SNPs in the pseudoautosomal region 1 of X-chromosome is similar to that for autosomes. One cautionary note, however, is that the total number of SNPs in PAR1 region is small and evolutionary dynamics of PAR regions differ from the rest of X-chromosome [29]. These limitations call for revisited analysis of PAR regions when more data become available.

## Discussion

The comprehensive analysis we conducted has found that the density of GWAS-detected SNPs on X-chromosome is 5-6-fold lower compared to that on autosomes. The difference cannot be explained by the overall lower density of SNPs on X-chromosome, a lower coverage of the X-chromosome by genotyping platforms, or an inferior call rate for X-chromosomal SNPs. Based on available publications [30,31] and our analysis (Fig 6), call rates for X-chromosome SNPs are similar to call rates for autosomal SNPs. The lower density of GWAS-detected SNPs on X-chromosome is not a result of a lower statistical power because female-restricted GWASs show the same level of the differences as all-phenotype GWASs.

Imputation is a commonly used approach to increase the number of SNPs analyzed in GWA studies [32,33]. Based on the results of a recent study [34], imputation accuracy is essentially the same for autosomal and X-chromosome SNPs: 89.3% versus 90.2. This suggests that imputation accuracy is unlikely to be the source of observed conspicuous differences in GWAS hit densities between X-chromosome and autosomes.

The other analytical technique frequently used in GWA studies is meta-analysis [35,36]. Meta-analysis improves the statistical power by combining statistics from several GWA studies. It is unlikely that this technique will contribute to X-chromosome versus autosomal differences in the GWAS hit densities since SNPs are included in meta-analysis regardless of their chromosomal location.

The other potential contributor to X-chromosome versus autosome differences in the density of GWAS findings is sex differences in allele frequencies. A recent study by Wang et al. [37] found that about 1% of SNPs on the X chromosome show sex differences in minor allele frequencies (MAFs). The study identified 2,039 SNPs with genome level differences in MAFs between males and females. We checked how many of those SNPs are reported by GWASs. We found that only one SNP from the list, rs4014653 was reported as a GWAS hit [38]. The frequency of GWAS-detected SNPs among SNPs with sex differences in MAF– 1/2039 is comparable with the fraction of GWAS-detected SNPs among all known SNPs (Fig 2). We estimated how many SNPs with sex differences in MAFs are on commonly used Illumina Humanomni5-4v1_c platform. We found that 21 SNPs are on the platform; therefore, the frequency of GWAS hits among directly genotyped SNPs is 1/21 = 0.05 which is similar to the overall frequency of GWAS hits on the platform, 0.08 (Fig 4). Thus it is unlikely that sex differences in MAFs is a major contributor to the deficit of X-chromosome SNPs in GWASs. X-chromosome SNPs do not show a higher deviation from HWE compared to autosomal SNPs. In fact, Graffelman et al. (2017) [39] found that the deviation from HWE is higher for autosomal SNPs. This finding, however, needs to be taken with caution because the study was based on the data from 1000 Genomes samples with a relatively low sequencing depth.

We hypothesized that the striking differences between X-chromosome and autosomes in the density of GWAS-detected SNPs is a result of a lower density of functional SNPs on the X-chromosome. Mutations on X-chromosome are more exposed to natural selection compared to autosomal mutations because of their hemizygosity in males and X-chromosome

inactivation in females [40]. As a result, both positive and negative selection is stronger on X-chromosome–faster-X hypothesis [3]. A number of studies provide a strong support for faster-X hypothesis [3,4,41]. Consistent with the idea of stronger negative selection on X-chromosome that "weeds out" functional variants from it, the study by Kukurba et al. [42] reported a significant depletion of expression quantitative trait loci (eQTL) on X-chromosome. Lower genetic variation on X-chromosome compared to autosomes was first reported by the SNP consortium [43] and later confirmed by other studies [44,45].

The analysis of proportions of segregating sites for the 7 categories of missense mutations stratified by the predicted effect on the protein function demonstrated that (i) the proportion of segregating sites increases as the predicted functional effect of missense mutations decreases for all chromosomes, and that (ii) the proportion of segregating sites for X-chromosome mutations is lower than the proportion of segregating sites for missense mutations in autosomal genes. The results provide support for the hypothesis of lower density of functional polymorphisms on X-chromosome as a result of stronger purifying selection.

Since the majority of *de novo* mutations are deleterious [46], purifying selection is the most prevalent [47] pressure on new variants. Purifying selection against deleterious mutations may change frequencies of the linked SNPs. This form of the linked selection is called background selection [48–50]. Background selection is widespread and significantly influences evolutionary dynamics of the linked SNPs regardless of their functional significance [50,51]. Therefore, changes in frequency of functional polymorphisms (e.g. missense mutations) usually change frequencies of linked nonfunctional SNPs (e.g. synonymous SNPs). A stronger selection against X-chromosomal mutations compared to autosomal ones is expected to result in a lower density of functional polymorphisms on X-chromosome. Neutral mutations are not subject to selection and, therefore, their frequencies will be similar on X-chromosome and autosomes. The results of our analysis are consistent with this expectation. First, the stronger X-chromosomal selection is expected to result in a lower density of SNPs on X-chromosome. That is exactly what we observed: the overall densities of SNPs from dbSNP, 1KG, and TOPMed databases were correspondingly 23%, 20% and 21% lower on X-chromosome compared to autosomes. Second, we found that the density of X-chromosomal SNPs located in transcribed regions is significantly lower compared to autosomes, while the densities of intergenic SNPs are similar for X-chromosome and autosomes. SNPs located in genic regions are more likely to be functional compared to intergenic [25,26] even though some of the intergenic SNPs also can be functional [52,53]. We also noted that the densities of both exonic and intronic SNPs on autosomes were higher compared to the corresponding densities on X-chromosome.

The results of the analysis of PAR1 region are also in line with the hypothesis that the deficit of functional SNPs on X-chromosome results in a deficit of GWAS-detected SNPs on X-chromosome. We found that in the PAR1 the fraction of GWAS-detected SNPs among all dbSNP variants was similar to that of autosomes. The finding provides further support to the hypothesis that the lower density of GWAS-detected SNPs on X-chromosome is a result of a stronger selection against functional SNPs located on X-chromosome resulting from the hemizogous state of X-chromosomal SNPs in males and X-inactivation in females.

SNP density in the PAR1 is more than 10 times lower than the SNP density in other parts of X-chromosome. This difference is surprising because meiotic recombination has been shown to be associated with a higher mutation rate [54] and the recombination rate in PAR1 region is 20-fold higher compared to the genome-wide average [2,55]. GC-biased gene conversion (gBGC) may provide an explanation of the lower SNP density in the PAR1. gBDC is a recombination-associated conversion—unidirectional transfer of genetic information from a 'donor' to an 'acceptor' sequence [56,57]. gBGC accelerates the fixation of guanine or cytosine alleles

and its effect is stronger than "mutagenic" effect of recombination [54]. gBGC is similar for functional and neutral polymorphisms because it acts regardless of the effects of the genetic polymorphisms on fitness.

In conclusion, we found that the density of GWAS-detected SNPs on X-chromosome is almost 6-fold lower compared to that on autosomes. Analysis stratified by SNP location in transcribed and intergenic regions supported the idea that the lower density of GWAS-detected SNPs on X-chromosome is a result of a stronger selection against functional SNPs on X-chromosome.

## Methods

### Data sources

Human Genome Assembly GRCh38.p13 was used to retrieve the sizes of individual chromosomes as well as the sizes of genic and intergenic regions. The dbSNP database Build 155 [58] was used to retrieve the number of SNPs in individual human chromosomes. To control for possible biases in the number of SNPs stemming from the targeted genotyping we used SNP data from two projects that used whole genome sequencing for SNP detection: 1000 Genomes Project phase 3 [59] and the TOPMed [60] project Freeze 5b. Phase 3 1000 Genomes Project has lower coverage of intergenic compared to genic regions [59]. Those differences are unlikely to influence the results of the analyses since proportions of exonic sequences are similar between X-chromosome and autosomes (S1 Table).

We used data from the Catalog of Published GWASs (CPGWAS) [61] to identify SNPs detected by GWAS studies. The catalog was accessed on June 12, 2022. For each chromosome we have identified the total number of SNPs reported by CPGWAS. SNPs detected and reported by multiple GWAS we counted as a single signal even when it was associated with different phenotypes or diseases. S1 Table shows the data on the chromosomal sizes including genic and intergenic regions, the number of SNPs on each chromosome and the number of GWAS-detected SNPs reported in CPGWAS for individual chromosomes. SNPs in the pseudoautosomal region one (PAR1) of the X-chromosome were analyzed separately because of its unique characteristics [29,62]. PAR2 region was not included in this analysis because it is too small and contains too few SNPs.

### Stratification by intergenic, genic, intronic and exonic regions

Data from the consensus coding sequence (CCDS) database (release 22) [63] were used to estimate the sizes of exonic, intronic and intergenic regions for individual chromosomes. First we retrieved the start and end exon and intron positions of each coding gene transcript. The sum of the lengths of all exons on a given chromosome was used as a size of the exonic region. We excluded overlapping gene sequences to make sure that each nucleotide was counted only once. The similar approach was used to estimate the total size of intronic regions. Noncoding genes were counted as part of the genic region. Start and end positions of noncoding genes were retrieved from the NCBI Gene database [64]. The size of the intergenic region was computed as the total size of the chromosome minus the combined size of exons, introns and noncoding genes. Genes located on the Y-chromosome and mitochondrial genes were excluded from the analysis.

### Possible explanations of the deficit of GWAS-detected SNPs on X-chromosome

We went through all technical/methodological explanations of the X-chromosome versus autosomal differences in the density of GWAS-detected SNPs of which we are aware. We

tested whether the following factors could explain the lower density of GWAS-detected SNPs on X-chromosome:

i. *Chromosome sizes.* To account for size differences we computed the number of GWAS-detected SNPs per megabase or kilobase of sequence for each chromosome and used this metric to compare X-chromosome and autosomes.

ii. *Overall SNP density.* Differences in the number of GWAS hits could be a result of differences in the overall SNP density between X-chromosome and autosomes. We estimated the densities of all SNPs reported in dbSNP, 1000 Genomes Project, and TOPMed databases and compared chromosomes by the number of GWAS-associated SNPs per thousand of all SNPs on a given chromosome.

iii. *Coverage by genotyping platforms.* To estimate the coverage of chromosomes by genotyping platforms we analyzed 28 most commonly used genotyping arrays [65]. For each genotyping array we estimated the number of SNPs for individual chromosomes. To account for differences in chromosomal sizes we counted the number of array SNPs per thousand nucleotides and compared X-chromosome and autosomes by this measure.

iv. *Call rates.* Because the amount of DNA available for genotyping X-chromosomal SNPs is halfthe amount compared to autosomal SNPs in men, one could hypothesize that the success rate for genotype calls is lower for X-chromosome SNPs compared to autosomal SNPs. We tested this hypothesis by estimating call rates in several commonly used genotyping platforms.

v. *Effective sample size.* In males the number of analyzed alleles for X-chromosomal SNPs is only half of that for autosomes, therefore, the effective sample size is lower for X-chromosomal SNPs. This may contribute to X-chromosome versus autosomal differences in GWAS hit densities. To address this issue we conducted an analysis of the data from female-only GWASs where effective sample size is the same for X-chromosome and autosomes.

vi. *Density of functional SNPs.* Theoretical and observational studies show that selection on the X-chromosome is stronger than on autosomes [3]. Selection directly affects functional mutations while neutral variants are free from selection pressure. GWAS signals are driven by the presence of functional/causal SNPs. If the density of functional variants is lower on the X-chromosome, the density of GWAS-detected SNPs is also expected to be lower. To test this hypothesis we compared X-chromosome and autosomes by the densities of genic (likely functional) and intergenic (unlikely functional) SNPs.

## Proportions of segregating sites for missense mutations with varying effect on protein function

We used Envision bioinformatics tool to predict the effect of missense mutations on protein function [66]. Envision combines experimental data on functional effects of missense mutations with a supervised, stochastic gradient boosting learning algorithm to quantify functional effects of missense mutations. Envision outperforms similar bioinformatics tools for prediction of functional effects [66]. Envision provides precomputed predictions for every possible single amino acid substitution in the human proteome. Among all possible amino acid substitutions (19 possible substations per amino acid) we identified the substitutions that can be produced by a single nucleotide substitution (SNS). For this we computationally mutated each single nucleotide in the coding sequence into 3 possible SNSs and identified those producing

missense mutations (see [67] for details). Possible missense mutations were stratified into 7 categories based on the Envision score–the lower score is associated with a stronger functional disturbance. Therefore, for each gene we estimated the number of potential sites for each category. We used 1000 Genomes Project data to identify how many of the potential sites exits as SNPs (S6 Table).

### Statistical analysis

We considered all autosomes as a group and tested the hypothesis that the X-chromosome belongs to this group. For each metric analyzed, e.g. the number of GWAS-detected SNPs per one million nucleotides, we computed the mean and standard deviation ($\sigma$) for autosomes. A standard Z-score was computed using the standard formula: $Z = \frac{x-\mu}{\sigma}$; where $Z$ is a standard Z-score, $x$ is the value of the metric for X-chromosome, $\mu$ –the mean value for autosomes, and $\sigma$ is standard deviation for the metric in autosomes. P-values corresponding to the given Z-scores were used for comparisons between X-chromosome and autosomes.

### Supporting information

**S1 Table. Chromosomal sizes, number of GWAS detected SNPs and number of SNPs reported in dbSNP, 1K genomes and TOPMed databases for individual chromosomes.**
(XLS)

**S2 Table. Number of SNPs on chromosomes across 28 commonly used genotyping platforms.**
(XLS)

**S3 Table. Number of GWAS-detected SNPs per 100 SNPs on genotyping platform.**
(XLS)

**S4 Table. List of female only GWAS.**
(XLS)

**S5 Table. Genome wide association studies randomly selected to find if they mentioned including males in HWE testing.**
(DOCX)

**S6 Table. Number of potential and segregating sites and the fraction of segregating sites per potential site for missense mutations stratified by the predicted effect on protein function (Envision Score).**
(XLS)

### Acknowledgments

We thank Mr. Spyridon Tsavachidis for help with analysis and Dr. Gorlova for discussions of the results.

### Author Contributions

**Conceptualization:** Christopher I. Amos.

**Formal analysis:** Ivan P. Gorlov.

**Funding acquisition:** Christopher I. Amos.

**Methodology:** Ivan P. Gorlov, Christopher I. Amos.

**Writing – original draft:** Ivan P. Gorlov.

**Writing – review & editing:** Ivan P. Gorlov, Christopher I. Amos.

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
