## [Decision Letter · Decision Letter 0]

6 Dec 2022

Dear Dr Gorlov,

Thank you very much for submitting your Research Article entitled 'Why are GWASs limp about the X chromosome?' to PLOS Genetics.

The manuscript was fully evaluated at the editorial level and by independent peer reviewers. The reviewers appreciated the attention to an important problem, but raised some substantial concerns about the current manuscript. Based on the reviews, we will not be able to accept this version of the manuscript, but we would be willing to review a much-revised version. We cannot, of course, promise publication at that time.

If you decide to revise the manuscript for further consideration at PLOS Genetics, please aim to resubmit within the next 60 days, unless it will take extra time to address the concerns of the reviewers, in which case we would appreciate an expected resubmission date by email to plosgenetics@plos.org.

We are sorry that we cannot be more positive about your manuscript at this stage. Please do not hesitate to contact us if you have any concerns or questions.

Yours sincerely,

Andreas Ziegler

Guest Editor

PLOS Genetics

Scott Williams

Section Editor

PLOS Genetics

Reviewer's Responses to Questions

**Comments to the Authors:**

Reviewer #1: The authors "found that the density of GWAS-detected SNPs on X-chromosome is almost 6-fold lower compared to that on autosomes. Analysis stratified by SNP location in

transcribed and intergenic regions supported the idea that the lower density of GWAS-detected SNPs on X-chromosome is a result of a stronger selection against functional

SNPs on X-chromosome." I quote this synopsis from the manuscript not only to summarize the work but also to show the authors' crisp and informative language.

The authors present a compelling case that few X-linked SNPs have been identified as being associated with traits in GWAS studies. Their datasets are large (1000 Genomes, TOPMed and the Catalog of Published GWASs) and their statistical analyses are simple but appropriate.

The authors state "The recommended approach to deal with this issue is to run HWE test only in females [21, 22]." This seems to contradict reference [21] where a test that combines HWE testing and male-female allele-frequency difference testing was discussed. The authors could confine their analyses here to female-only studies, but they could report on the rates of HWE rejection on autosomes vs X-chromosome.

The authors focus on GWAS-detected SNPs. What do they conclude about findings from other methods, such as linkage studies? Would those not also be affected by selection against X-variants?

The authors' "We tested all possible technical/methodological explanations of the X-chromosome versus autosomal differences in the density of GWAS-detected SNPs." should at least be modified to refer to explanations they can think of or have read about. It does not detract from their work to acknowledge there still may be other technical/methodological explanations. It doesn't appear that they looked at male-female allele frequency differences, for example. Have they ruled out possible bias in CPGWAS resulting from fewer reported GWAS that looked at X-chromosome SNPs?

Reviewer #2: This is an interesting manuscript which provides some data to suggest that the paucity of GWAS signals on X chromosome go beyond previously described phenomenon.

The title is potentially stigmatizing: specifically, the word limp has common sexual inferences: erectile dysfunction, limp-wristed, that I suspect the authors probably do not mean. An alternative word choice might be more appropriate.

I think that the SNP consortium was the first to show reduced variation on X cf autosomes: https://www.nature.com/articles/35057149

Please specify which build of dbSNP was used. Similar for which release of TOPMed.

I don’t quite understand what was done for Fig 4. (Table S3). Did they look at each GWAS with X chrom signal, determine which platform was used, and then compute the # of X vs autosomal signals. Or did they just take all GWAS signals and see whether the SNP was on specific platforms?

Most GWASes now use imputation and meta-analysis where component studies have genotyped different arrays. How do these two aspects impact these findings? This is a major topic that is not addressed anywhere in the manuscript and could have implications for a number of their observations. In addition, commonly used references such as the 1000 Genomes phase 3 data has recently been shown to have sex differences in allele frequencies for some SNPs, which potentially could impact the results of analysis: PMID: 35639794.

It is more conventional to refer to 1000 Genomes Project rather than ‘Thousand genomes’. Which version did they use? Seems to be phase 3. The latest data is high coverage WGS. Cell 2022: PMID: 36055201. Is there a specific reason that they did not use that?

Have they examined the allele frequency distribution of variants on the X vs autosomes in dbSNP?

In addition to GWAS of female-specific traits, there have been some GWAS of traits in sex-specific fashion (e.g. WHR). Could this add useful information? How do you know that these female-specific GWASes even analysed the X chromosome?

Regarding call rate, there is a typo in the ‘Affymetrix platRform’ in Fig 6. Can the authors be specific about which Affy platforms, since some had very high rates of variants being cleaned due to various QC issues, e.g. https://www.ncbi.nlm.nih.gov/pmc/articles/PMC4072456/ Also, they do not provide X chr call rates stratified by sex. This is important cince call rate is a common QC metric.

Regarding HWE, there is a more important issue in females on X that one generation of random mating is not sufficient to produce HWE in females – it can require many more generations. I think Graffelman & Weir discussed this (Heredity, 2016) who cite Kimura and Crow 1970. Is there any information about what % of SNPs are failing HWE on X vs autosomes?

For the analysis of SNP density (Intergenic/intronic/exonic), the authors should acknowledge that phase 3 1000 genomes used high coverage WES (including many different heterogeneous capture and sequencing technologies), but only low coverage WGS (in addition to two GWAS arrays). It would be interesting to see if this difference exists for the recent high coverage WGS from 1000 Genomes (see above). In addition, since there are both synonymous and non-synonymous exonic variants, do they see differences in density between these groups? It would be helpful to label the X axes of the Fig 7 by chromosome number. What happens when they look in ClinVar variants, does that support fewer bioinformatically predicted functional variants on X genes than autosomes?

Is the gene density similar on X vs autosomes?

‘Twice lower’ would be better worded as half’.

When they argue that for a dominant model the sample size is small for males may be true, but most analysis assumes additive model. What is the implication on power then?

Analysis of density across the X chromosome assumes that there are no gaps in the sequence. My recollection is that particularly in PAR1 there are multiple large gaps in GRCh37, which typically would result in low SNP density and poor imputation (if even attempted or possible). What happens when they take such gaps into account? Do they see reflections of these gaps when they look at the density of SNPs on different arrays? A recent paper examined population genetics of PARs and may contain relevant information: PMID:33872316.

I have a similar comment related to gaps in sequence in other chromosomes. For example the first ~ 5Mb of chr 21 has N in GRCh18. Did they count this to the genic region in Tables S1?

How were sequential meta-analysis of the same trait dealt with in the analysis. For example there are 268 studies of BMI in the GWAS catalog today. Presumably most found the loci with largest effect (e.g. FTO). Are they counting these as independent or a single signal?

**Have all data underlying the figures and results presented in the manuscript been provided?**

Reviewer #1: Yes

Reviewer #2: **No: **They say they reviewed 20 random GWASes, but do not provide the references for them.

PLOS authors have the option to publish the peer review history of their article (what does this mean?). If published, this will include your full peer review and any attached files.

Reviewer #1: No

Reviewer #2: No

---

## [Decision Letter · Decision Letter 1]

28 Jan 2023

Dear Dr Gorlov,

Thank you very much for submitting your Research Article entitled 'Why does the X chromosome lag behind autosomes in GWAS findings?' to PLOS Genetics.

The manuscript was fully evaluated at the editorial level and by independent peer reviewers. The reviewers appreciated the attention to an important topic but identified some concerns that we ask you address in a revised manuscript.

We therefore ask you to modify the manuscript according to the review recommendations. Your revisions should address the specific points made by each reviewer.

Yours sincerely,

Andreas Ziegler

Guest Editor

PLOS Genetics

Scott Williams

Section Editor

PLOS Genetics

Reviewer's Responses to Questions

**Comments to the Authors:**

Reviewer #1: The authors have addressed my concerns.

Reviewer #2: In general, the authors provide adequate responses to my previous comments. However, they do not provide any justification for their use of the phase 3 1000 genomes data rather than the high coverage data (https://www.sciencedirect.com/science/article/pii/S0092867422009916). In particular, there were many different capture and sequencing technologies used for the exome sequencing in phase 3, that could in part be driving the apparent paucity of bioinformatically predicted functional variants on X (which of course tend to be rare and are most impacted by technical errors). This is puzzling, since the high coverage data is of much higher quality. Recently, the high-coverage data has been re-aligned to the T2T-CHM13 consortium build, in a Science paper entitled ‘A complete reference genome IMPROVES analysis of human genetic variation‘ (https://www.science.org/doi/10.1126/science.abl3533), making it even more puzzling why they persist with the phase 3 data.

Fig 7. Still this figure has ‘one thousand genome SNPs’ as title. Also, TopMED should be TOPMed.

In the paragraph before where Fig 6 is referenced in the results ‘and stratified by gender (Figure 6).’ I think they mean by sex.

The authors should describe the limitations of using data from dbSNP since it is a user- submitted database. For example it has been shown that conventional bioinformatic approaches to variant calling on the sex chromosomes results in failure to call thousands of variants per subject (PMID: 31289836), especially in the PARs, and it is my impression that few (or non) of the consortia that submit large numbers of SNPs to dbSNP have employed this modified method. The authors make no mention of this, and the problem is potentially even worse for 1000 genomes phase 3 data, since I remember that they performed an ensemble variant calling across ~20 different methods, and although I have not examined each and every method, I am skeptical that any of them employed X-specific variant calling.

The authors respond to the HWE question, by referring to Graffelman 2016 Fig 7. This figure shows qq plots for females and males for chi2 and exact tests. There are no direct comparisons for results across SNPs, so it is not correct for the authors to make any statement that ‘For some SNPs the results … are different,… but overall are not substantial’. Although Graffelman 2016 state the inflation of the chi2 is due to rare alleles, there is no data shown to support this conclusion. Further, in the Discussion when the authors cite Graffelman 2017 comparing autosomes to X for HWE, they should be clear that these data come again from 1000 Genomes phase 3, from two very small populations, and the limitations of which I have described above (e.g. mean depth=6).

When the authors analyse TOPMed data, they should consider the nature of the cohorts that contribute to TOPMed, and the bias that this may introduce. As I understand this is an NHLBI-funded program, including classic CVD cohorts. I have never seen the age distribution of TOPMed as a whole, but I suspect it’s relatively old. Could loss of X diversity due to premature mortality from XLR diseases result in part of the apparent deficit of X exonic variants. Is there a birth cohort that could be used for comparison purposes? For example, variants in COL4A5 (Xq22.3) are a common cause of Alport syndrome, that classically presents in childhood/early adulthood in males. Individuals with Alport syndrome due to variants in this gene typically develop end-stage kidney disease in their 30s and likely either are not alive to participate in cohorts like those in TOPMed, or are too sick to want to participate. It would be useful to look at common severe XLR disease genes (FRAX, DMD, BMD,. hemophilia A, G6PD) to see whether the diversity in those genes differs from other X genes. Similar considerations likely apply to 1000 Genomes samples: I presume that only adults were approached to be included.

**Have all data underlying the figures and results presented in the manuscript been provided?**

Reviewer #1: Yes

Reviewer #2: Yes

PLOS authors have the option to publish the peer review history of their article (what does this mean?). If published, this will include your full peer review and any attached files.

Reviewer #1: No

Reviewer #2: No

---

## [Editor Report · Decision Letter 2]

15 Feb 2023

Dear Dr Gorlov,

We are pleased to inform you that your manuscript entitled "Why does the X chromosome lag behind autosomes in GWAS findings?" has been editorially accepted for publication in PLOS Genetics. Congratulations!

Yours sincerely,

Andreas Ziegler

Guest Editor

PLOS Genetics

Scott Williams

Section Editor

PLOS Genetics

Comments from the reviewers (if applicable):

**Data Deposition**

http://datadryad.org/submit?journalID=pgenetics&manu=PGENETICS-D-22-01162R2

**Press Queries**

---

## [Editor Report · Acceptance letter]

22 Feb 2023

PGENETICS-D-22-01162R2 

Why does the X chromosome lag behind autosomes in GWAS findings? 

Dear Dr Gorlov, 

We are pleased to inform you that your manuscript entitled "Why does the X chromosome lag behind autosomes in GWAS findings?" has been formally accepted for publication in PLOS Genetics! Your manuscript is now with our production department and you will be notified of the publication date in due course.

With kind regards,

Zsofia Freund

PLOS Genetics

On behalf of:
